# The Role of Cognition in Dishonest Behavior

**DOI:** 10.3390/brainsci13030394

**Published:** 2023-02-24

**Authors:** Adrián Muñoz García, Beatriz Gil-Gómez de Liaño, David Pascual-Ezama

**Affiliations:** 1Department of Social Psychology and Methods, Autonomous University of Madrid, 28049 Madrid, Spain; 2Department of Medical Specialties and Public Health, Rey Juan Carlos University of Madrid, 28933 Móstoles, Spain; 3Department of Financial Management and Accounting, Complutense University of Madrid, 28040 Madrid, Spain

**Keywords:** dishonesty, cognition, dishonesty models, deception, inhibition, working memory, creativity, social cognition

## Abstract

Dishonesty has received increased attention from many professionals in recent years for its relevance in many social areas such as finance and psychology, among others. Understanding the mechanisms underlying dishonesty and the channels in which dishonesty operates could enable the detection and even prevention of dishonest behavior. However, the study of dishonesty is a challenging endeavor; dishonesty is a complex behavior because it imposes a psychological and cognitive burden. The study of this burden has fostered a new research trend that focuses on cognition’s role in dishonesty. This paper reviews the theoretical aspects of how such cognitive processes modulate dishonest behavior. We will pay special attention to executive functions such as inhibitory processes, working memory, or set-shifting that may modulate the decision to be (dis)honest. We also account for some frameworks in cognitive and social psychology that may help understand dishonesty, such as the Theory of Mind, the role of creative processes, and discourse analyses within language studies. Finally, we will discuss some specific cognitive-based models that integrate cognitive mechanisms to explain dishonesty. We show that cognition and dishonest behavior are firmly related and that there are several important milestones to reach in the future to advance the understanding of dishonesty in our society.

## 1. Introduction

Dishonesty has been extensively studied over the past decade, as it reflects a significant interest in human behavior and has a considerable impact on professional structures such as companies and organizations [1]. Many studies have focused on individual differences in committing dishonest behavior and the personal traits associated with dishonesty [2,3,4]. However, recent novel experimental paradigms have facilitated a new trend of research focused on the mechanisms underlying dishonesty, including the cognitive processes directly associated with it.

By breaking down the concept of dishonesty (e.g., “*We need to suppress the truth when we lie*”, or “*We got to monitor whether the receptor of our scheme is suspecting something*”), elements and terms related to the cognitive literature, such as suppression or monitoring, can be easily identified. Such decompositions have led to a new body of research that assumes that dishonesty has a cognitive disadvantage/burden over honesty. Several works have emerged to test for the extra resources demanded by dishonest behavior [5]. One of the first studies suggesting that dishonesty, specifically verbal lying, is cognitively effortful was that of Zuckerman et al. [6]. They showed higher reaction times and specific physiological indexes related to attentional processes for lying compared to truthful scenarios. They suggested that deceitful speech must comply with another’s knowledge and purposes (external consistency) and must maintain one’s own internal consistency between verbal and behavioral cues. Subsequently, several works reported interesting results on the potential role of cognition in dishonest behavior, which we will review below. Thus, the present paper aimed to review what we know about cognitive processes in dishonest behavior.

Despite the extensive evidence of cognition mediating dishonest behavior, a consensus on how cognition modulates dishonesty is still lacking in the literature. This disagreement may be due to several factors. First, there is the complexity of both concepts: cognition and dishonesty. Cognition comprises several processes, such as memory, language, learning, perception, or executive functions (EF), among others. Dishonesty is a complex phenomenon, as well. Some of the most frequent forms of dishonesty are cheating, lying, fabrication, sabotage, impersonation, etc. Concepts such as deception, which includes omitting information, misreporting, or misleading messages, among other similar behaviors, are also part of what we understand as dishonest behavior while not strictly being considered as lies. Recently, Srour and Py [7] proposed a classification of the ways in which people produce deceptive messages on a daily basis. They called it the Elementary Deception Modes (EDM), and it is based on a personal three-year introspection study. In Table 1, we can see definitions and examples of some of the most relevant terms described throughout the literature.

Second, recent research has shown that dishonest behavior should not be dichotomized as honest/dishonest. Dishonesty seems to be within a continuum or “grey scale” of different levels of dishonest behaviors [8,9]. Traditional methods for studying dishonesty that infer dishonesty at the aggregate level might have oversimplified the way we understand it. Recent novel paradigms [9] have allowed for individual-level analysis, showing different types of dishonesty that can be modulated or even maximized [10]. We will showcase these latest breakthroughs to understand how cognition can mediate on several types of dishonesty.

Hence, we have structured the present review while trying to answer the following questions: Which are the main cognitive processes related to dishonesty? How does social cognition modulate dishonest behavior? How are other cognitive processes related to deception? Which cognitive models can help us to understand dishonesty?

According to the literature, executive functions (EF) are the most significant cognitive processes for understanding dishonest behavior. Most research studying cognition and dishonesty has focused on EF. Thus, we will first examine the most important studies on EF and dishonesty, specifically on inhibition, working memory, and task switching—three of the most relevant cognitive processes defining EF. Although we will discuss at some points how these cognitive processes are implemented in our neural structures and circuits, our focus will be on the cognitive processes themselves, paying less attention to structural brain implementation or connectivity. There are other works for readers interested in an extensive review of the neuroscience of dishonesty [11,12]. While we will cite some of those works in this review in the need to understand how cognition mediates dishonesty, our aim was not to describe the cerebral structures and circuits involved in dishonesty but to explore and understand how cognitive processes mediate different types of dishonest behavior.

Dishonesty is essentially a social behavior. It depends on others’ perceptions and how they interact with our world understanding. Therefore, a critical question is how dishonesty may interact with social cognition. The cognitive Theory of Mind (ToM) is the ability to understand others’ beliefs. Thus, it seems to be essential to understanding dishonesty. Furthermore, although discussing how personality traits can explain dishonest behavior is beyond the scope of this review (and would warrant a separate review in itself), we will talk about some related factors, such as creativity. We will review some interesting work on dishonesty and creativity, as well as other work reviewing the role of language in dishonest behavior.

Finally, we will review several cognitive models that can provide insights into understanding dishonesty and how the different cognitive processes previously described are interconnected. Figure 1 provides a brief overview of the main themes that we will discuss in the following sections. After each section, we will examine the findings in detail and their potential relevance to understanding the various aspects of dishonesty.

To summarize, as the field of dishonesty continues to evolve, this review aimed to explore the literature and examine some of the key insights and major breakthroughs made thus far in understanding the relationship between cognition and dishonesty. We will finish the review by discussing limitations and the new challenges to come.

## 2. Which Are the Main Cognitive Processes Related to Dishonesty?

Dishonesty is exhausting. Those who have tried to uphold a lie for a long time likely understand how exhausting and cognitively demanding it can be. Studying the relationship between cognition and dishonesty has become critical to understanding this demanding and complex behavior. As previously mentioned, one of the earliest works to suggest this relationship was Zuckerman et al. [6], which examined the detection of dishonest speeches. Dishonesty detection is one of the fields that has contributed most to the study of cognition in dishonesty.

Vrij [13] notes the shift in the literature on lie-detection research from seeking physiological differences between truth-tellers and liars to the importance of examining the various cognitive processes and strategies employed when being dishonest. Traditionally, techniques for detecting dishonesty were based on physiological measures such as increasing the interviewee’s arousal in interrogation settings. However, since honest individuals may also exhibit high activation levels due to the stress of being under interview pressure, efforts have shifted to studying the cognitive strategies and mechanisms underlying dishonesty. Many current techniques for detecting dishonesty support a cognitive approach and show greater accuracy than traditional methods [14]. These strategies are often based on depleting the resources of the dishonest individual with the assumption that cognitive resources are necessary to sustain a lie. Therefore, cognitive processes such as attention, working memory, attentional shift, inhibition, and others appear to play a crucial role in the study of dishonesty: dishonesty frequently pursues a goal (usually for self-serving purposes) that must be maintained and updated (working memory), be flexible enough to adapt to the context, and focused by attentive processes (selective attention, inhibition, shifts, etc.). These processes are commonly referred to as executive functions (EF), which include goal setting, cognitive flexibility, and attentional control [15]. According to Hofmann et al. [16], EF processes consist of three essential tasks: working memory (WM) operations, such as the maintenance and updating of relevant information (‘updating’), inhibition of prepotent impulses (‘inhibition’), and mental set-shifting (‘shifting’). We will review works studying each of those processes and how they interact in dishonest behavior.

### 2.1. Working Memory (WM)

In the study of dishonesty and working memory (WM), most researchers in the field have employed dual-task paradigms to investigate how dishonesty is affected by increased cognitive loads. The underlying assumption is that maintaining a lie or engaging in deceptive behavior requires additional cognitive effort, and this should be reflected in changes in behavior and performance under increased WM load conditions. For example, Vrij et al. [17] found that manipulating cognitive load by asking participants to recall a story in forward and backward order revealed more behavioral cues that facilitate the detection of deception-like behavior. Other studies have used similar techniques, such as asking interviewees to stare at interviewers or perform multiple tasks simultaneously [5,18]. These studies suggest that increased cognitive load impedes the lying process. However, it is unclear whether this means that individuals require more cognitive resources to lie or that those with higher cognitive workloads are more likely to lie.

To address this question, Van’t Veer et al. [19] used the classic “die-under-the-cup” paradigm along with a WM task. Their aim was to determine whether individuals under high cognitive loads would be more or less likely to engage in (dis)honest behavior, hypothesizing that if individuals have limited working memory resources (due to other information loads), they would have more difficulty committing dishonest behavior. Participants rolled a die while maintaining a set of words (more or fewer words). They had to decide which die outcome to report, knowing that depending on the outcome reported, there would be higher or lower rewards. In this paradigm, participants were not supervised and could report any outcome, whether it was the one that they actually obtained or a different one that could give them a higher reward. The results showed that participants under low cognitive load were significantly more dishonest than their counterparts, suggesting that cognitive load impairs dishonesty.

In a recent study, Speer et al. [20] showed that neural correlates of cognitive-control demands in a Stroop task could be useful to understand dishonest behavior. However, they found that cognitive control can promote both honesty and dishonesty, depending on one’s moral default.

Further evidence on the impact of working memory (WM) on dishonesty is derived from studies on child development. These studies are particularly intriguing because executive functions are not fully developed during childhood. In fact, depending on their age, differences in WM capacity can play a crucial role [21]. Alloway and colleagues [22] investigated whether WM (verbal and visuospatial) was linked to successful lying in a temptation-resistance game among children aged 6–7. The children were placed in a room where they could see the answer to a question an examiner would ask soon. They were specifically instructed not to look at the answer. Results showed that “good liars” (children who saw the “forbidden” answer and lied about it) had higher scores in verbal working memory compared to “bad liars” (those who saw the answer but did not lie about breaking the rule). However, they found no differences in visuospatial WM capacity between the two groups. Other studies have reported similar findings supporting the role of WM in children’s dishonesty [23,24].

Therefore, WM load appears to influence or moderate dishonesty. New theoretical models including WM as an explanatory factor suggest that WM’s role in dishonesty can be twofold: it can come from the need to retrieve information required for the task, but also to update new information, items, and targets as the behavior evolves [25]. In this process of information/activation, truth and untruth are retrieved and activated, balancing the maintenance and inhibition of relevant information for the task. Hence, the role of inhibition seems to be critical for explaining dishonest behavior too.

### 2.2. Inhibition

Deliberate actions require cognitive control to overcome instinctive responses [26]. This principle can also apply to dishonest behavior, where the truth is understood as the instinctive preeminent response. In situations where telling the truth is the natural response and lying is the deliberate behavior, mental resources are needed to suppress the dominant response of telling the truth [27]. Inhibition can take many forms, such as resisting distractions, prepotent responses, or resistance to proactive interference [28]. However, research on dishonesty has usually focused on whether people tend to suppress the predominant response of truth. Even a simple question like “What is your name?” can trigger an automatic response (one’s actual name) that must be overridden to provide a different answer (a potential lie). Experimental data support dishonesty as placing a cognitive burden on individuals by requiring them to suppress the dominant response (the truth). Debey et al. [29] used the Sheffield Lie Test to demonstrate that lying increases response times, as it necessitates truth-suppression and attention to inhibiting the instinctive response. In a subsequent study, Debey and colleagues [30] found that individuals with better inhibitory skills were better at lying, as measured by their “lie effect” score (response time for lying minus response time for telling the truth).

However, other researchers using different types of inhibitory tasks have not found significant differences between lying and telling the truth. Fenn et al. [31] reported a facilitation effect when participants in a high inhibitory load group were perceived as more trustworthy than those in a low inhibitory load group after inducing them with increased demands of inhibitory resources. These contradictory results could be explained by the methodological differences in the studies. As Caudek et al. [32] suggested, Go/No-go or Stop signal tasks could provide a more reliable way to measure inhibition of prepotent responses in lying. These two tasks are typical inhibitory tasks (compared, for instance, with the Sheffield Lie Test) where participants must inhibit a response to a given stimulus and respond to others. They suggested that the stop-signal tasks are preferable, as they provide more precise measures of stopping latency compared to other inhibitory tasks.

In their study, they first measured inhibition with a stop-signal task, and in another session, participants performed a Sheffield Lie Test. However, they did not replicate the previous findings and found no differences between conditions of lying and telling the truth, nor any relation with the function of the stop-signal task score among individuals. Aïte et al. [33] attempted to address the previous conflicting findings and hypothesized that basing the study on dual-process theories might help. To test their hypothesis, they designed a behavioral task (the Pirate Task) built upon a negative-priming paradigm [34]. They reported an interesting negative-priming effect and longer RTs in the deceptive trials, supporting the cognitively demanding effect of lying. The authors concluded that inhibition plays a key role in dishonesty, supporting dishonesty as being more cognitively challenging than honesty.

However, more research is needed to understand the fundamental mechanisms of inhibitory processes in dishonesty. The study of task switching (set-shifting), which involves changing the mental set between tasks, could provide more clues about the functioning of inhibitory processes in dishonesty.

### 2.3. Set-Shifting

Dishonesty is not always a clear-cut binary or radical action; there are several levels at which dishonest behavior can occur. For instance, one can choose to not divulge the complete truth in a given situation, resulting in the omission of relevant information, but not as a direct lie. Furthermore, not all acts of dishonesty are completely dishonest. In Pascual-Ezama et al.’s study [9], participants exhibited a range of behaviors while performing the typical “die-under-the-cup” task. These behaviors varied from radical actions, such as not rolling the die at all and reporting the outcome for the maximum reward, to rolling the die multiple times to receive a moderate to maximum reward. There were also various shades of gray in between these conditions, such as not rolling the die but not reporting an outcome for the maximum reward, rolling the die until the maximum reward was achieved, or rolling the die just once, but lying about the outcome to receive more money. Thus, different levels of dishonest behavior can result in situations where one may alternate between truthful and dishonest behavior. However, such situations can create an extra cognitive load, as one must make an extra effort to maintain the coherence of their dishonest speech or behavior [35].

In a meta-analysis, Christ et al. [36] demonstrated that brain regions associated with attentional set-shifting overlap with those activated during fraudulent activities. Many of the studies included in the meta-analysis used the Sheffield Task, which measures the switching cost between truth and lying by manipulating the proportion of truth/lie trials within the task. Van Bockstaele et al. [37] tested whether changing the percentage of switches between telling the truth or deceiving trials in the Sheffield Lie Test affected the proportion of the general truth effect. They found that increasing the number of times participants were asked to lie in the Sheffield test did not affect response times on the switch costs, but differences showed up for errors. Participants with more deceptive trials made fewer errors, likely due to their “deceptive continuity”.

Similarly, Foerster et al. [38] studied how different mindsets or attentional states might modulate truth-telling. They replicated switch costs effects under dishonest behavior and also examined the “dishonest intention effect” by manipulating participants’ awareness of the following trial type (dishonest vs. honest). They found that response times were shorter when the next trial type was shown in advance, reducing switch cost effects by anticipating the trial type. Additionally, switch costs from “dishonest-to-honest” were higher than “honest-to-dishonest” when participants anticipated the trial type. Pfeuffer et al. [39] also found similar results in studies using interviewers who manipulated honest/dishonest questions and attempted to switch between questions to exhaust the cognitive resources of the interviewers.

These studies provide evidence for the cognitive burden of dishonesty and highlight the importance of studying executive functions in dishonest settings to understand and potentially prevent cheating, lying, and other forms of dishonesty. However, as dishonesty is a complex behavior, it cannot be fully understood by a single approach. Recent models of dishonesty have considered executive functions and other important processes, such as social cognition and creativity, to explain the mechanisms of dishonesty. The social nature and transgressive character of unethical behavior have led to linking dishonesty to paradigms like the Theory of Mind, with executive functions playing a role in these theories [40,41]. In the following section, we will examine the main findings in social cognition that contribute to understanding dishonest behavior from a cognitive perspective and attempt to answer the question of how social cognition can account for understanding dishonest behavior.

## 3. How Is Social Cognition Related to Dishonesty?

Dishonesty has a social dimension, which, with its cognitive nature, makes social cognition processes crucial for understanding it. Dishonest behavior often occurs in a social context and thus requires an understanding of both the context and the intentions of the individuals involved [42]. Models of dishonesty, including social cognition components [43], support the idea that the ability to understand, infer, and anticipate others’ thoughts (Theory of Mind—ToM) is essential for comprehending dishonest behavior. Before lying or cheating, it is crucial to determine what the other person expects you to do or say. Moreover, before being dishonest, it is necessary to know (or infer) what the other person already knows [43]. The act of lying, cheating, or deceiving always involves someone else, making dishonest behavior by definition a social act between at least two people. Therefore, social cognition is believed to be involved in dishonesty, but more importantly, the well-known ToM must be developed to engage in any form of dishonest behavior. Lisofsky et al. [44] gathered evidence from 22 fMRI and 2 PET studies in a meta-analysis, investigating the influence of social cognitive processes on dishonesty. Among other things, one of the main findings of this meta-analysis was that different cognitive processes (aside from executive functions) related to social interactions and representations, such as moral reasoning and ToM, might be involved in dishonesty. Many models of dishonesty that we will discuss below have integrated ToM as an essential component for explaining dishonest behavior [7].

ToM is the ability to reason about others’ mental states and beliefs [45]. It usually appears between 3–4 years old in the typically developing population [46] and it is thought to be a prerequisite of deception [47]. Ding et al. [48] conducted an experiment to study whether 3-year-old children who previously showed no lie capacity could increase their disposition to lie in a hide-and-seek task when trained in the ToM. Results showed an increase in lying in the experimental group compared to the control group (also 3-year-old children) not trained in ToM. Similar studies with children have replicated Ding et al.’s results [49,50], showing that ToM is necessary to commit any dishonest behavior. However, Talwar and Lee [51] did not find differences in the likelihood of lying between 3-year-old and older children in a temptation-resistance paradigm game. Interestingly, what they did report was an improvement of the “skill” in lying (maintaining consistent statements) as a function of the age (see also [52]).

There are also studies of the ToM and dishonesty with adults. Apperly et al. [53] studied brain-injured adults’ performance on a non-inferential false-belief ToM task. Participants were presented with two sentences and one picture. One sentence was about a man’s beliefs, and the second was a real fact. The participant’s task was to judge whether the image faithfully recreated the sentence. Results showed a higher cognitive burden when subjects were informed about a false belief conflicting with the fact. Similarly, El Haj et al. [47] studied how ToM could mediate between destination memory and dishonesty using a sample of healthy adults. They found that participants were more likely to remember false information if they knew that the person who provided it was highly skilled at reading emotions and mental states. Other studies in the academic-cheating field have proposed that sometimes, cheating may be mediated by different types of teachers or professors, supporting teacher mood states as having a possible impact on their student’s potentially dishonest behavior [54].

From another perspective, Johnson et al. [55] carried out a study with adults to test whether self-awareness was related to dishonesty. The authors defined self-awareness as the ability to perceive our feelings or others’ self-perception [56,57]. That is, self-awareness capacity seems to depend on a previous acquisition of ToM [58]. They recorded various scenes interpreted by different actors. The scripts that actors had to perform could contain truthful biographic information, exaggerations of desirable characteristics, or negative traits. Before recording the tapes, the actors filled out the self-conscious scale [57], which measures self-awareness. Then, participants watched the recordings and were asked to judge whether the actors were honest or dishonest. Results showed that those actors with higher private self-awareness performance were more trusted and worthier. This supports the idea that social cognition abilities (like self-awareness and ToM) may lead to better dishonest skills.

Thus, empirical evidence supports ToM as being necessary to perform dishonest behavior. However, some questions remain open to understanding how social cognition, together with other cognitive processes, works in dishonesty.

## 4. How Are Other Cognition Processes Related to Dishonesty?

The literature reviewed so far has emphasized the significance of executive functions in conjunction with social development to explain dishonest behavior. However, there may be other cognitive processes to consider to better comprehend dishonest behavior, such as creativity and language. The well-known Zuckerman’s model analyzes dishonest discourses and compares the paraverbal elements in language between honest and dishonest behaviors to detect dishonesty. The literature has identified several significant differences in the production of dishonest speech, including shorter length, more pauses and hesitations, more errors, and vaguer details [59,60] (especially when there is a complex lie to “maintain” [61]), although well-premeditated dishonest speeches can be quite detailed [62] (even if they often include fewer contextual details [63]). Honest speeches also tend to be more positive than dishonest ones [64]. In addition, a study using the die-under-the-cup paradigm revealed that native speakers were significantly more dishonest than non-native speakers [65]. Therefore, exploring language and discourse in humans provides a valuable source for understanding the interrelation between cognition and dishonesty, and more research is needed in the future, particularly in how language and discourse interact with attentional and executive functions to better depict and comprehend dishonest actions.

On the other hand, numerous studies have explored the relationship between creativity and dishonesty. Creativity is a multifaceted concept encompassing various cognitive functions and can be approached from various perspectives, such as from the result of a new creation in any field [66], to a cognitive process where motivation plays a crucial role in the creative process [67], or even as a type of personality trait [68]. However, regardless of the perspective, it is generally agreed that creativity is something innovative and different from the regular and typical. That is, creativity seems to imply breaking some sort of “rules” [69]: this is where the critical link with dishonesty lies. Dishonest behavior typically involves acting against regular and normative principles and may include some creative process when lying.

From this point of view, Gino and Ariely [70] conducted studies exploring the link between creativity and dishonesty. Their first study investigated whether participants with higher creativity levels were more likely to cheat in a quiz report. They also measured intelligence, as it has often been linked to creativity [71]. The results showed that participants with higher levels of creativity were more likely to cheat than those with lower levels of creativity. Regression analysis showed a significant relationship between dispositional creativity and dishonesty when intelligence was controlled. However, there was no evidence of a relationship between intelligence and dishonesty or creativity (see [72] for another perspective). In another experiment, they tested whether a “creativity prime” could increase cheating when reporting results from a problem-solving task, where participants earned money based on their performance. Participants first completed a scrambled-sentence test [73], which involved creating sentences from randomly positioned words. For the experimental group (creatively primed), 12 out of 20 sentences contained creativity-related words. Later, they performed a 20-matrix task (problem-solving task), where they were asked to find two numbers in each matrix, composed of 12 three-digit numbers that add up to 10. A cheating index was easily calculated by comparing the real performance and their reported performance. The results showed significant differences between the two groups, where creative priming elicited more dishonest behavior.

Finally, in a third experiment, Gino and Ariely [70] studied whether a “wide range of justification” would allow people to engage in dishonest behavior. Previous findings suggest that creativity would increase one’s capacity to justify unethical behavior in a self-serving way [74,75]. They conducted an experiment using the creative priming (creativity prime vs. non-prime) and justification (high vs. low) as factors. The justification factor was manipulated using the “die-under-the-cup” task. One group was asked to roll the dice once (low justification), whereas the other group could roll the dice several times before reporting the outcome (high justification). The authors reported that when the dice could be rolled more than once (high justification), the proportion of payoff results was higher compared to the “just once roll” condition (low justification). Furthermore, those participants that were “creatively primed” reported higher payoff results than those in the non-prime condition. The results also showed a significant interaction between both factors, providing evidence to support the hypothesis that creativity and self-serving bias are indeed related to increased cheating. Similarly, Gino and Wiltermuth [69] investigated whether dishonesty leads to better creative results in a problem-solving task (rather than testing whether creativity enhances dishonesty in the same problem-solving task). Indeed, the results showed that the “cheater” group performed significantly better in both the problem-solving and creativity tasks.

Additional research has also connected creativity and dishonesty [76,77,78], further strengthening the evidence of their relationship and leading to similar conclusions. Creativity might facilitate deceptive responses by priming them or operating through a self-serving bias in both directions. Another explanation could be that creativity may be interpreted as the ability to generate justifications to lessen the conflict of being dishonest. In either case, dishonest behavior involves certain levels of creativity, likely because both concepts involve breaking rules.

## 5. How Do Cognitive Processes Interact with Each Other in Dishonesty? Dishonesty Theoretical Models Accounting for Cognition

After reviewing the most significant cognitive processes involved in dishonest tasks, it is important to put them into a theoretical and conceptual framework to understand how they interact to explain dishonest behavior. In this last section, we will review theoretical models that consider cognition as a critical factor underlying dishonest behavior. Our aim is to highlight how cognition can be crucial to understanding dishonesty within an explanatory model of dishonest behavior.

Zuckerman’s model [6] was likely the first important model incorporating cognition as a key component explaining dishonesty. The rest of the models reviewed here have been somewhat inspired or motivated by Zuckerman’s model. Drawing on earlier work by Ekman and Friesen [79], Zuckerman et al. postulated four essential assumptions to describe and understand deception. First, deceivers attempt not to be perceived as dishonest by monitoring their verbal and non-verbal behavior. Second, deception entails an increase in arousal or physiological activation. Third, deception raises emotions such as guilt and anxiety. Finally, deception is cognitively complex due to the high demands of consistency, plausibility, and coherence with the receiver’s knowledge.

Although the original model lacks some deep explanations for these principles, these premises imply a clear assumption of cognitive mechanisms’ involvement in explaining dishonesty. Working memory is clearly noticeable in the first and last assumptions (monitoring one’s behavior and high cognitive demands). Language is also explicit in the first assumption, as it particularly applies to verbal and non-verbal behavior. Emotions considered in the third assumption can perfectly imply social aspects, as they involve others to be guilty with, for instance, and probably ways to control self-representations and self-awareness. The increase in arousal postulated in the second assumption could also be related to attentional and inhibitory aspects, as they imply focusing on something to avoid something else. Thus, in general, the model proposes an interrelation of different aspects explaining dishonesty that involves several of the cognitive processes described in the previous lines of this review.

The Interpersonal Deception Theory (IDT) [80] posits that deception involves a cognitive effortful communicative process. According to this theory (based on Zuckerman et al.’s model), lying requires strategic reasoning, self-representation, monitoring, control of verbal or non-verbal cues, and analysis of the context. The IDT focuses on contextual requirements and the cognitive resources needed to manipulate them. For example, it introduces the “Truth-Bias”, which suggests that we tend to assume that others are mostly honest [81]. The authors propose that we expect honesty due to the automatic activation of truth, although they interpret social rules as a precursor to this principle. However, specific contexts, such as a police debriefing or academic pressure, may break that bias and promote cheating. Thus, detecting deception would vary based on the receiver’s previous expectations and contextual variables, which are associated with the social nature of dishonesty and can inspire new research on cognitive biases in interpreting our and others’ desires and needs during social interactions.

Similarly, the Truth-Default Theory (TDT) [82] uses contextual factors and Zuckerman’s insights to present a deception model that accounts for the specificities of the context where deception occurs. The TDT, like the IDT, assumes that most people are honest, but a given context can prompt dishonest behavior. The model also incorporates concepts such as truth-bias and truth-default. Truth-bias refers to the current tendency to believe that others are honest despite their level of honesty. In contrast, truth-default denotes the unconscious belief that honesty is the default rule in communication. This model includes consciousness/unconsciousness aspects that could lead to new research on (un)consciousness in cognitive science and neuroscience. According to the TDT, the truth-default principle is not broken until there is no contextual index, triggering suspicion and different levels of (un)conscious cognitive processing.

Another relevant model in the field is the Activation–Decision–Construction–Action Theory (ADCAT) proposed by Walczyk et al. [43]. This model integrates executive function, social cognition, decision making, and emotional cognitive components to develop a comprehensive model of high-stakes dishonesty. The model consists of four components: activation, decision, construction, and action. The first component, activation, involves the determination of whether an important truth is required by the context or the interlocutor’s intentions. This insight is stored in working memory. The Theory of Mind (ToM) is also needed for this phase. In parallel, useful information is retrieved from long-term memory (LTM) if it already exists, and it is also stored in working memory. Then, the decision component takes place through a goal-optimization strategy guided by previous experiences and emotions, which are typically socially based. This sets the motivation to lie, cheat, or commit any dishonest act (or not), depending on the context and previous emotional experiences. However, the decision strategy is not always rational, and heuristics guide it under “extreme” situations (e.g., lying/cheating if a particular grade on an exam, like the GRE, is necessary to be accepted into the preferred college). The third component is construction, which involves the deceptive manipulation of information. Networks of semantic, episodic, and emotional memory are activated based on the information gathered in the previous steps, helping to follow the plausibility principle and/or to adjust the lie to others’ knowledge. Finally, the action component comes into play, and the truth is inhibited so that the deception is delivered. At this level, the behavior is also monitored and measured based on new requirements (changing situations) and other reactions, considering deception as an active process, as in the Interpersonal Deception Theory (IDT2), which we subsequently review.

The Information Manipulation Theory 2 (IMT2) [35] is a variation of the ADCAT and TDT models. Unlike the TDT, which focuses on contextual factors to explain deception, the IMT2 is a deceptive discourse production model that mainly concentrates on language aspects. The IMT2 assumes that deception and truth share the same discourse system, a premise inherited from the ADCAT model, in contrast to the other models, which are more grounded in Zuckerman’s model. Another difference with traditional models is that the IMT2 assumes that statements involve parallel processing. Therefore, deceptive communication follows a top-down serial process, rather than a continuous, simultaneous production, revision, and maintenance process. Although the procedures may differ, cognitive control is still considered a critical factor in explaining dishonest behavior. As such, memory, attention, decision-making, and task-switching processes are all thought to be involved in deceptive communication. The IMT2 pays particular attention to information treatment but considers deception an active process that needs to be continuously readapted based on information fluctuations.

In contrast to previous models that focused on contextual and discourse approaches, Sporer [83] proposed a deception model based on Baddeley’s and Hitch’s working memory model [84]. This working memory model of deception raises two main ideas based on previous studies on cognitive load manipulations. First, our central executive or attentional control is limited in maintaining information. Second, attentional control plays a key role in generating and supporting both lies and truth [74]. Sporer’s model integrates a wide range of processes related to deception, such as memory, speech production, and attention. The critical point is that their mechanisms rely mainly on the central executive or attentional control described in Baddeley’s working memory model. Importantly, Sporer also highlights the central executive function as a critical factor in linking active information and long-term memory retrieval to verbal production and behavioral control in the act of committing dishonest behavior.

Moreover, the “schemata” concept raised in several models is critical in Sporer’s model. According to the schemata approach [85], sequences of experiences are stored as scripts. However, we do not literally save episodes, but only a summary of these episodes. When a script is repeated, all the consistent information becomes a more generic schema, and irrelevant details become more likely to be forgotten. There are some deviations from the general schema; schema-inconsistent information sometimes increases the likelihood of being firmly stored and rehearsed in the future due to its particularity or extraordinariness. Applied to the construction of lies, the schemata theory implies that truthful recalls will be more detailed because of the schema-inconsistent information. Lies will be “poorer” because they are based on general schemas. Therefore, Sporer’s [84] working-memory-based model to explain dishonesty is especially interesting because of its powerful cognitive-based explanation of complex lie production. Based on their assumptions, written reports of events are more cognitively demanding than spoken descriptions, consistent with the literature reviewed in the previous sections. These reports increase the likelihood of reporting more precise details rather than more challenging ones.

Similarly, Lane and Wegner [86] proposed the “Secrecy” model to study dishonest behavior from a cognitive perspective. According to their theory, secrecy involves the suppression or inhibition of the secret itself. However, attempting to suppress the secret also triggers the intrusion of the secret, creating a paradoxical cycle of suppression. In other words, if one tries “not to think of a polar bear,” an image of a polar bear will suddenly come to mind. Wegner’s iconic process, inspired by a phrase from Dostoevsky, has been useful in treating obsessive disorders and in illustrating the functioning of their secrecy model. Similarly, the secrecy model uses this iconic process to explain how keeping secrets works. The explanatory mechanism follows a dual-process theory, similar to Aïte et al.’s [33] reasoning model. The secret automatically comes to mind (type 1), and then one must deliberately monitor and suppress it (type 2) in order to succeed. In other words, lying (e.g., keeping relevant secret information) requires cognitive effort to suppress the truth. Finally, it is worth mentioning a recent proposal model based on a disruptive new framework explaining deception from a different point of view. The previous models essentially come from the study of deceptive discourses. However, the General Theory of Deception (GTD) [7] defends the multifaceted nature of deception, dividing it into three steps: planning, execution, and cover-up. The planning stage involves the selection of the target, the construction of the lie, and the identification of potential obstacles. The execution stage involves delivering the lie, while the cover-up stage involves maintaining the lie and preventing detection. All stages involve diverse cognitive processes here reviewed (working memory, inhibition, attention shifting, etc.). Although this identification of stages sounds similar to what ADCAT proposed, GTD gives more importance to individual differences and contextual requirements. Additionally, it is presented not only to explain high-stakes demeanors like ADCAT but also to provide a wider picture of deception. One of the main contributions of this model is its classification of different deceptive behaviors. They identified 99 Elementary Deception Modes (EDMs), which are daily examples of deceptive behaviors. Another interesting proposal is the way they modeled one’s end-to-end deceptive behavior. They proposed a five-factor model (benefits, punishment, risk, execution, dissonance) for which the expected outcome of the dishonest behavior is evaluated, followed by a decision–performance algorithm that describes, with specific cognitive mechanisms, all the mental and behavioral analysis and execution of the deception. The construction of a detailed algorithm will enable future research to empirically falsify the model, and more importantly, it will integrate new dishonest paradigms with its conception of dishonesty as a wide-ranging and complex behavior, as has been shown in recent empirical studies [4,9].

## 6. Final Conclusions

This review of evidence on dishonesty that has been collected over the past few decades shows the significance of cognitive processes explaining and understanding dishonest behavior. A key assumption underlying the link between dishonesty and cognition is the effort required to sustain dishonest behavior in our cognitive system. This assumption is backed by empirical data on the need to suppress automatic honest responses, the constant monitoring of one’s behavior and discourse while being dishonest, and the evaluation of context and inferences about other people’s involvement. All these assumptions support a theoretical framework based on human capacity limitations, where dishonesty is a complex behavior that demands constant activation and deactivation of information and requires sufficient resource availability to occur [87].

However, as we have seen, there are some contradictory results among studies. As in many other research fields, these differences could result from the diversity of manipulations and paradigms among studies. More research is needed to disentangle the puzzle of how cognition modulates dishonesty. However, despite the variations among models and studies, there are some common agreements. Most suggest that dishonesty is a process that can be broken down into stages that involve monitoring (working memory), inhibition, and set-shifting of information. The specifics of how this occurs vary between models and studies and the empirical data supporting each of them. For instance, certain models postulate that specific general cognitive processes, such as inhibition, might affect behavior differently depending on the dishonest task manipulation or contextual variables [30,31]. Inhibition may impair dishonest decision making by controlling impulsive actions, but on the contrary, it may act as a facilitator during dishonesty under certain contextual situations. Parceling out cognitive processes and their involvement in different stages of dishonesty could help develop more robust explanatory models, such as the one proposed by Walczyk et al. [43] or the General Theory of Deception [7].

It is crucial to note that incorporating new experimental tasks may enhance our understanding of how cognition operates in dishonest behavior. Pascual-Ezama et al. [9] developed a new online version of the “die-under-the-cup” task, which enables individual tracking of dishonest behavior, unlike the aggregated results of previous studies. Results from this study, as well as others replicating the findings [4,88], are critical to studying dishonesty and understanding its potential variability at different levels and scales. By taking an individual approach, researchers can uncover a diverse set of dishonest profiles based on different strategies, allowing us to differentiate between the nature and gradient of dishonesty (e.g., cheating, lying, full-extent dishonesty, etc.). This approach and new models like the GTA [7] open a new avenue for studying the cognitive mechanisms that underlie these distinctive dishonest profiles. Including new methods in the study of dishonesty could help researchers understand the specific mechanisms involved in the entire process of dishonest behavior. Further studies should emerge to answer important questions that still need to be addressed in the field. In fact, as shown in Figure 2, there has been a significant increase in the number of papers that explore the relationship between cognition and dishonesty over the past few years. This trend highlights the importance of studying how cognition operates in different forms of dishonest behavior.

However, although the present work has a significant contribution and strength in its multidisciplinary approach to addressing cognition involvement in dishonesty, that also brings some limitations. Given dishonest behavior’s complex and extensive nature, we aimed to review most breakthroughs using diverse experimental paradigms. The paradigms reviewed here are summarized in Table 2. As the table shows, they can be very different, responding to various questions about the involvement of cognition in dishonesty. As we have pointed out before, some discrepancies in the literature reviewed here have arisen. These discrepancies could generate the impression of contradictory results and some confusion, although they can respond to differences among the paradigms. Despite these limitations, the empirical evidence clearly supports cognition’s essential role in understanding dishonest behavior. We encourage researchers in the field to consider the new methods and advances presented in some recent results and theories to help design further studies and experiments in the future to address these limitations.

In conclusion, the growing research interest in the cognitive mechanisms underlying dishonest behavior has resulted in valuable insights into how dishonesty operates. These advances in scientific understanding may even pave the way for detecting and preventing dishonest behavior. A key aim of studying dishonesty is to find ways to mitigate its impact on professional structures, companies, organizations, politics, and daily human interactions. The investigation of cognition in dishonesty can help us achieve this goal.

## Figures and Tables

**Figure 1 brainsci-13-00394-f001:**
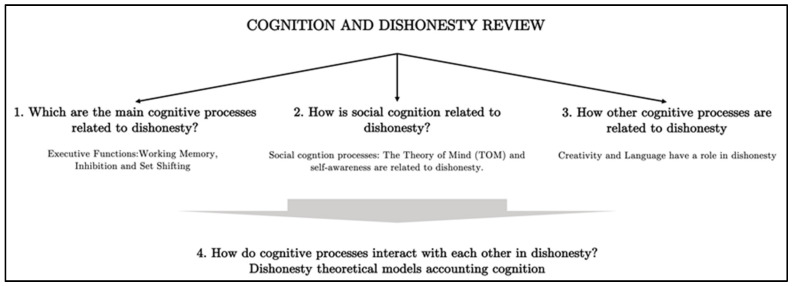
Cognition and Dishonesty Review Structure.

**Figure 2 brainsci-13-00394-f002:**
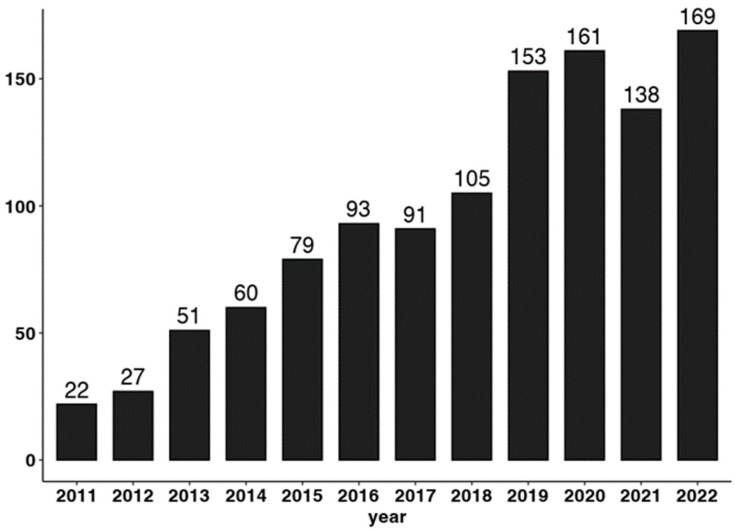
Publications about dishonesty and cognition found on the Web of Science.

**Table 1 brainsci-13-00394-t001:** Types of dishonest behavior (simple definitions) and examples.

Label	Simple Definition	Example
Cheating	Breaking the rules	Tricky dice
Lying	Not telling the truth	Saying you are younger than you are
Deception	Hiding all the truth or avoiding information	Dishonest excuse for a deadline extension
Fabrication	Falsification of data	Manipulating experiment results
Sabotage	Not permitting others to do the work	Deflating an opponent’s tire in a race
Impersonation	Paying someone else to replace you	Someone else does an exam for you
Bribery	Giving money for something	Paying to skip the queue
Plagiarism	Using other information without attribution	Copying someone else’s poem

**Table 2 brainsci-13-00394-t002:** Most commonly used experimental tasks to measure dishonesty and reviewed in the present work.

	Classic Die-under-the-Cup	Pascual-Ezama et al. [9] Die-under-the-Cup	Sheffield Lie Test	Matrix Task	Pirate Task	Spot the Difference Task	Sender–Receiver Tasks
*Level of Analysis*	Aggregate	Individual	Individual	Individual	Individual	Individual	Individual
*Multiple Dishonesty Measures*	No	Yes	No	No	No	No	No
*Performance-Related*	No	No	No	Yes	No	Yes	No
*Deliberate Dishonesty*	Yes	Yes	No	Yes	No	No	Yes
*Allows Profit-Maximization*	Yes	Yes	No	Yes	No	Yes	Yes
*Payoff Competition*	No	No	No	No/Yes	No	No/Yes	Yes

## Data Availability

Not applicable.

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
