# Peer review of "The Role of Cognition in Dishonest Behavior"

_brainsci, 2023, doi:10.3390/brainsci13030394_

Round 1

Reviewer 1 Report

This paper is interesting and ambitious. It is interesting because it provides a relatively exhaustive survey of the scientific literature on the cognitive processes involved in dishonest behaviour, particularly in lying. It is ambitious because its main originality is to situate the question on a much more general level than lying. The paper also addresses a wide range of dishonest behaviour: in addition to lying & deception, cheating, fabrication, sabotage, impersonation, bribery, plagiarism.
Incidentally, a better distinction should be made between lying and deception, which are generally used as synonyms.
It is this broad spectrum that makes the paper so interesting, but it is also this broad spectrum that leads to weaknesses in the argument. Indeed, the vast majority of the literature reviewed concerns the study of lying. As a result, the paper's common thread that cognitive processes are the key variable in accounting for dishonest behaviour is based on studies of lying. Even within this framework, the authors do not fully cover the literature. In particular, The General Theory of Deception (https://doi.org/10.1037/rev0000389), which is admittedly recent, is not discussed, although it does nuance the importance of the cognitive processes at work in the production of lies.
As for other types of dishonest behaviour, there is a lack of careful consideration of the importance of cognitive processes. For example, why would cheating on an exam (copying from another candidate, or using unauthorised sources) generate a higher cognitive cost? It seems easy to argue that it rather implies a certain cognitive laziness. The same goes for the idea of impersonation or bribery, or plagiarism.
I am more convinced by the authors' recourse to Theory of Mind, which is probably very transversal to all dishonest behaviour. The notion of creativity addressed by the authors is also interesting and probably also potentially transversal to all dishonest behaviour.
I therefore think that the authors should take their paper back to the level they claim to address, i.e. all dishonest behaviour, which is the main originality of the paper. Otherwise, the scope of the paper should be restricted to lying.
One detail: The benefit of recalling an event in reverse order is not the most convincing strategy for Increasing cognitive load to facilitate lie detection. The only case in which the reverse order strategy produces the desired effects is when the interview starts with this strategy, which is certainly not usable in the field. On the other hand, the use of drawing or schematics is effective and allows reasoning in terms of cognitive load.

Author Response

First of all, thank you for your review. We believe the changes have improved the original paper, so thank you very much for the review process. 

First, we have incorporated the General Theory of Deception (GMS) into our manuscript (pp 11-12). We also believe it is a significant contribution to the scope and focus of our work, and needed to be included. We appreciate your suggestion.

Second, we have also acknowledged your concern about the blurred distinction between deception and lying (see pp2). We have included a more precise definition and have restructured that section to make it clearer and less ambiguous.

Finally, we have taken into account the rest of the suggestions from the other reviewers, including a new paragraph with some limitations of the review (pp13), together with a deep proofreading of English throughout the whole manuscript. We have corrected spelling errors and restructured some sentences and paragraphs to enhance readability and understanding.

Thanks for your feedback, we believe your suggestions have improved this new version of the manuscript.

Reviewer 2 Report

The article is very well written and discusses a relevant behavioral topic. I have the following comments for a minor revision:

1) There are quite a few current behavioral literature based on the socio-economic connection of brain functionalities and a possible connection towards the dishonest behavior. I urge the authors to consider this element in their review.

2) A brief limitation and strength section need to be incoroporated for this review.

Author Response

First of all, thank you for your review. We believe the changes have improved the original paper, so thank you very much for the review process.

In response to your suggestions, we have added a new paragraph in page 13 about some limitations of the review, including also a new table (Table 2, pp14). We appreciate your suggestion as we believe it has improved the clarity of the review and part of the message we wanted to transmit.

Additionally, we have taken note of your comments about the literature on neural correlates and dishonesty. We have included valuable references to extensive meta-analytic works throughout the text to support our argument. Although we believe that neuroimaging studies provide a strong foundation for our thesis, we recognize that this topic may be beyond the scope of our work, and we have chosen to focus on the functional processes of cognition. We have also included a clarification to this point and reefer the interested reader to other reviews about brain functioning in dishonesty (see end of page 2).

Furthermore, we have carefully reviewed and improved the quality of the English language used in the manuscript by proofreading it. We have corrected spelling errors, restructured some sentences and paragraphs, and enhanced the readability and natural flow of the text.

We are grateful for your constructive feedback and valuable contributions to our manuscript.

Reviewer 3 Report

The paper refers to an important research topic and it is well-positioned in Journal scope. From a methodical and practical point of view, the investigated problem is interesting. The structure of the document is transparent. The article is interesting.

In the article, the Authors raise a significant research problem, which is the understanding the mechanisms underlying dishonesty.

Dishonesty is a complex behavior and cannot be fully understood by taking a single approach. Therefore, the authors analyze various aspects, seem to play a key role in the study of dishonesty. Importantly, the authors also recognize the limitations of the review.

In this analysis, they consider the impact of cognitive processes, such as attention, working memory, attention shift, inhibition, etc. (executive functions) and other factors, such as social cognition, creativity and language. They also review theoretical models that recognize cognition as a factor in dishonest behavior.

The topics and scope reviewed are very extensive.

The article is written correctly in terms of structure, however, from the point of view of the quality requirements for the scientific study, some question arise:

Is the review of such a vast subject complete enough? 75 references per review article is not much.

For example:

- “There is evidence showing that lying also depends on the linguistic context. …” – 1 literature reference !!!

- “Section 3. How is social cognition related to dishonesty?” - only 12 references to literature.

- etc.

Please respond to this comment regarding the completeness of the review. I believe that, at least in some parts, the review should be expanded to include additional literature citations.

Author Response

First of all, thank you for your review. We believe the changes have improved the original paper, so thank you very much for the review process.

Regarding the number of references, we agree that some statements lacked proper referencing, and we have taken steps to rectify this issue (there are now about 90 references that seems more appropriate for a review, even though the field is still new). We carefully examined the entire manuscript and added more references, particularly in the social cognition section, which we believe has clearly improved the quality of our work. Thanks for pointing it out.

We have also reviewed the quality of the English language used in the manuscript by proofreading it. We have made necessary corrections to spelling errors and restructured some sentences and paragraphs to enhance readability.

Finally, we have taken into account the rest of the suggestions from the other reviewers, including a new paragraph with some limitations of the review (pp13), and other corrections explained above.

Thanks for your feedback too, as we believe they have improved this new version of the manuscript.

Round 2

Reviewer 1 Report

The authors have carried out extensive revision work. I am not sure that the revision fully addresses my concern that the literature review covers all dishonest behaviour in a balanced way. However, this may be considered to be due to the very state of the scientific literature on these issues.
The paper presents an original opening for the consideration of dishonest behaviour in general (and not only lying) and the review of the cognitive processes involved in this behaviour constitutes a scientific contribution.  I expect that this paper will stimulate research on dishonest behaviour.